# Lipidomics Reveals Elevated Plasmalogens in Women with Obesity Who Develop Preeclampsia

**DOI:** 10.3390/jcm12082970

**Published:** 2023-04-19

**Authors:** Ian M. Williams, Matthew E. Albertolle, Alexander J. Layden, Sunny Y. Tao, Susan J. Fisher, Robin E. Gandley, James M. Roberts

**Affiliations:** 1Lipid Nanoparticle Platform Department, Takeda Development Center Americas, San Diego, CA 92121, USA; 2Drug Metabolism and Pharmacokinetics Department, Takeda Development Center Americas, San Diego, CA 92121, USA; 3Department of Epidemiology, University of Pittsburgh Graduate School of Public Health, Pittsburgh, PA 15261, USA; 4University of Pittsburgh Medical Center and Division of Gastroenterology, Hepatology and Nutrition, University of Pittsburgh, Pittsburgh, PA 15213, USA; 5Department of Obstetrics, Gynecology, and Reproductive Sciences, University of California San Francisco, San Francisco, CA 94143, USA; 6Magee-Womens Research Institute, Pittsburgh, PA 15213, USA; 7Department of Obstetrics, Gynecology and Reproductive Sciences, School of Medicine, University of Pittsburgh, Pittsburgh, PA 15213, USA

**Keywords:** lipids, maternal obesity, pregnancy

## Abstract

**Objective**: Preeclampsia (PE) is a prevalent pregnancy disorder worldwide with limited preventative treatments available. Obesity triples the risk for PE, yet only 10% of women with obesity develop PE. The factors that distinguish PE from uncomplicated pregnancies in the context of obesity have not been fully established. **Methods**: We studied a cohort of women with obesity throughout pregnancy to identify lipid mediators and/or biomarkers of PE. Blood samples were collected at each trimester and analyzed by both targeted lipidomics and standard lipid panels. Individual lipid species were compared by PE status at each trimester, as well as by self-identified race (Black vs. White) and fetal sex. **Results**: Standard lipid panels and clinical measurements revealed few differences between PE and uncomplicated pregnancies. Targeted lipidomics, however, identified plasmalogen, phosphatidylethanolamine, and free fatty acid species that were elevated in the third trimester of women with PE. Furthermore, race and trimester of pregnancy were considerable sources of plasma lipidomic variation in women with obesity. **Conclusions**: First and second trimester individual plasma lipid species do not predict the development of PE in obese women. In the third trimester, PE patients have elevated levels of plasmalogens—a class of lipoprotein-associated phospholipids that have been implicated in the response to oxidative stress.

## 1. Introduction

Preeclampsia (PE) is a common pregnancy disorder (2.7–8.2% of all pregnancies) that is a major cause of maternal and infant mortality worldwide [1,2,3]. The disorder usually manifests in the last two months of pregnancy, becoming more frequent as pregnancy approaches term (≥37 weeks of gestation). Once established, little can be done to modify disease progression except for indicated delivery which terminates the pathophysiology but can also add to infant morbidity and mortality [4]. Other than aspirin [5,6] and calcium supplementation [7], there is a lack of preventative interventions which directly target PE pathophysiology. A major challenge in developing such interventions is that PE pathophysiology is heterogeneous [8] and depends on the maternal environment in which the fetus develops. Studying PE in more homogeneous risk-prone populations may allow us to identify early drivers of PE pathophysiology and therapeutic targets for PE prevention.

One maternal environment that increases risk for PE is a state of obesity. Obesity has been positively correlated with PE in several populations [3,9,10,11,12]. In general, there is an approximately three-fold increased risk in women with a BMI greater than 30 compared to women with BMI of 21 or less [9]. Given the 3–5% rate of PE in the general population, this indicates that approximately 10% of women with obesity develop PE whereas the majority (90%) do not [13]. While obesity is complex, it most likely presents a more homogenous pathophysiology than PE in the general population. Thus, we sought to determine what distinguishes the 10% of women with obesity who develop PE from those who have uncomplicated pregnancies. 

Few studies have investigated the differences between women with obesity who develop PE and those who have uncomplicated pregnancies. In previous studies of pregnant women with obesity, the only major clinical biomarkers associated with PE were a family history of thrombotic disease, low placental growth factor, increased uterine artery resistance, and elevated mean arterial pressure [14,15]. Notably, neither HDL, LDL, nor TGs were associated with PE development in these studies. It has been shown, however, that changes in individual lipid species, independent of broad lipid classes, can identify individuals at risk for cardiovascular disease [16]. Furthermore, Lee et al. found several elevated phospholipid species in a cohort of lean women that predicted PE [17]. Therefore, we asked whether early-pregnancy differences in individual lipid species could distinguish the 10% of women with obesity who develop PE from the 90% who do not. 

In this study, we performed a longitudinal survey of the plasma lipidome in a cohort of pregnant women with obesity. We utilized liquid chromatography-mass spectrometry (LC-MS)-based targeted lipidomics [18] to measure a diverse array of individual lipid species. The primary objectives of this study were to (1) identify early pregnancy lipid biomarkers of PE in obese women, and (2) uncover potential lipid-based mechanisms of PE pathogenesis. We also performed secondary analyses on the effects of race and fetal sex on plasma lipid species. This allowed us to determine (a) whether lipid drivers of PE may differ by race and (b) whether the increased risk of preterm PE conferred by female fetuses [19] is associated with changes in lipid species.

## 2. Materials and Methods

### 2.1. Study Population

In this nested case-control study, patients were identified from the Pregnancy Exposures and Preeclampsia Prevention 3 (PEPP3) Study, a prospective cohort study conducted to examine factors that may predispose women with obesity to PE. The PEPP3 cohort consisted of 600 women with obesity who delivered at Magee-Womens Hospital (MWH) in Pittsburgh, Pennsylvania between 2009 and 2014. Patients were recruited early in pregnancy (between 6 and 16 weeks’ gestation) from the prenatal care clinics of MWH and through the Women’s and Infants’ Registry. Women with multiple gestations and those with pre-existing hypertension, diabetes, or major medical conditions were excluded. The PEPP3 study was approved by the University of Pittsburgh Institutional Review Board (IRB #PRO08050339). Written informed consent was obtained from all study participants.

The present sub-study included only women with pre-pregnancy BMI ≥ 30.0 kg/m^2^, as determined by self-reported pre-pregnancy weight and height measured at first visit, who delivered live singleton infants. In total, 33 women with obesity who developed PE were identified from the cohort and were group-matched by BMI (+/−3 kg/m^2^), race, parity, gestational age at sampling (+/−1 week), and smoking status to 67 women with obesity who had normal pregnancy outcomes. Approximately 2/3 of the women in the study self-identified as African American and 1/3 identified as Caucasian.

### 2.2. Preeclampsia Definition

Using the criteria defined by the American College of Obstetrics and Gynecology, [20], preeclampsia was defined as systolic blood pressure (BP) > 140 mmHg and/or diastolic BP > 90 mmHg, and proteinuria > 0.3 g/day (corresponding to 1+ or greater on urine dipstick or >0.3 protein/creatinine ratio) after 20 weeks’ gestation in a previously normotensive woman. Blood pressure was determined by averaging five measurements taken at the time of hospital admission for delivery, prior to administration of any medications that would alter blood pressure. Proteinuria was determined as more than 0.3 g of protein in a 24-h urine collection, 2 or more grams of protein in a random urine sample, 1 or more grams of protein in a catheterized urine sample measured by dipstick, or a protein/creatinine ratio greater than 0.3.

### 2.3. Sample Collection

Demographic and anthropometric data and samples of maternal blood were obtained at the initial visit between gestational weeks 6–12. Blood pressure during each study visit was calculated as the average of three measures from a validated automated device with the appropriate size blood pressure cuff based upon measured arm circumference. Non-fasting blood samples from each patient were collected at subsequent second (18–20 weeks) and third (34–36 weeks) visits, and plasma was stored at −80 °C until the time of batch analysis. Of note, there were no differences in the time since last food between groups at any trimester (Appendix A–C). All samples were collected prior to the development of PE. Any samples that were collected within one week of delivery in women with PE were excluded, as metabolic variables measured from these samples may represent overt PE. Gestational age was determined by the last reported menstrual period and validated by early ultrasound.

### 2.4. Lipid Concentrations

Maternal serum samples were analyzed in duplicate for triglycerides (interassay coefficient of variation CV = 4.6%), non-esterified fatty acids (CV = 5.1%), and total cholesterol (CV = 5.4%), using commercially available colorimetric detection kits (Pointe Scientific). HDL cholesterol was measured by using a PEG reagent (Pointe Scientific) to precipitate the LDL and VLDL fractions from the sample, with the remaining supernatant assayed with the colorimetric cholesterol detection kit to determine HDL cholesterol concentrations (CV = 12.5%). Serum concentrations of LDL cholesterol were calculated from HDL cholesterol, total cholesterol, and triglyceride measurements using the Friedewald equation.

### 2.5. Lipidomics

Lipids were first extracted using a modified Bligh and Dyer procedure which utilizes dichloromethane instead of chloroform. Deuterated internal standards, including 54 different lipid species covering 10 lipid classes, were then added to each extract to allow for quantitative determination of lipid concentrations. These standards have diverse lipid head groups, acyl chain lengths and degrees of unsaturation which span the fragmentation and ionization patterns of the individual lipid species [21]. 

Samples were then analyzed on the Lipidyzer platform [18] which consists of a Nexera X2 UHPLC system connected to a Sciex QTRAP 5500 mass spectrometer equipped with a SelexION differential mobility spectrometry (DMS) cell (Sciex). Each sample was analyzed in two separate runs: one with the DMS cell on and one with it off. Targeted lipid analytes were measured by multiple reaction monitoring in both positive and negative ion mode. Lipidomics Workflow Manager software was used for automated data acquisition, processing, and reporting. A total of 460 individual lipid species were detected in plasma samples which were analyzed over 5 separate run days. Lipids were only included for further analysis if they were detected on at least 4 of the 5 run days. This resulted in data on 385 individual lipid species being used for downstream statistical analysis (see below).

### 2.6. Statistical Analysis of Clinical Data

Descriptive statistics were used to examine baseline maternal characteristics, which are reported as mean +/− standard deviation for continuous variables and percentages for categorical variables. Potential differences in baseline characteristics of PE cases and uncomplicated pregnancy controls were evaluated using *t*-tests and chi-squared tests for continuous and categorical variables, respectively. Linear mixed models with a random intercept to account for within-pregnancy variation in lipid concentrations were performed using the *lme4* package in R to compare concentrations of lipids across trimester between women with PE and uncomplicated pregnancy controls. Differences in lipid concentration trends by PE status were tested by interaction between the effects of trimester and PE. Linear mixed models were likewise used to compare lipid concentrations by fetal sex and maternal race across trimester. Model residuals and random intercepts were graphically represented by histograms and quantile–quantile plots to assess normality assumptions of linear mixed models. For any violations, lipid concentrations were natural log-transformed and model residuals and random intercepts were reassessed for normality. Scatter plots of fitted values vs. model residuals were used to assess for violations of homoscedasticity. An alpha level of 0.05 was assumed for nominal significance and a Bonferroni-corrected alpha was used for significance after multiple comparisons of lipids. For any significant differences in lipids between groups throughout pregnancy (e.g., PE status, race, or fetal sex), post hoc pairwise *t*-tests were performed at each trimester.

### 2.7. Broad Lipid Classes from Lipidomic Data

To calculate the concentrations of broad lipid classes (e.g., diacylglycerols) for Figure 1, the concentrations of individual lipid species within each class were summed. These data were then analyzed by linear mixed effects modeling in Prism. Data were log-transformed prior to model fitting to improve normality of the residuals. Models included group, trimester and interaction fixed effects, and a random intercept to account for repeated measures on individual patients. We used the Greenhouse–Geisser correction to adjust p-values for lack of sphericity. Šidák’s multiple comparisons test were then used to compare lipid concentrations between groups at each time point.

### 2.8. Lipidomic Statistical Analysis

#### 2.8.1. Pre-Processing of Lipidomics Data

All lipidomic statistical analyses were performed using Prism (Graphpad) and various packages in R. To prepare data for statistical analysis, we handled missing values with a two-step approach. First, we removed individual lipid species if they were undetected in more than 20% of the samples in both groups at each time point [22]. This resulted in excluding 105 lipid species and left 280 species for downstream analysis. Any remaining missing values were imputed using k nearest neighbor averaging [23] with the *impute* R package.

#### 2.8.2. Univariate Analysis

To identify individual lipids which differ between groups, we first performed Mann–Whitney U tests on each lipid species at each time point during pregnancy. To correct *p*-values for multiple comparisons, we controlled the false discovery rate with the Benjamini–Hochberg procedure using the *fdrtool* (version 1.2.17) R package. Lipids with an adjusted *p*-value < 0.1 were considered for subsequent analysis. For each of the lipids identified by this procedure, we used data from both groups and each time point for linear mixed-effects modeling as described above in Broad lipid classes.

#### 2.8.3. Principal Component Analysis

Initial exploratory principal component analysis (PCA) was performed using various functions in the *PCAtools* (version 2.10.0) and *mixOmics* (version 6.18.1) [24] R packages. Data were first log-transformed, centered, and scaled. We then performed multilevel PCA using *mixOmics*. In this approach, PCA is performed on the within-subject variation in order to take advantage of the repeated measurements.

#### 2.8.4. Sparse Partial Least Squares Discriminant Analysis

Sparse partial least squares discriminant analysis (sPLS-DA) was used to build models for predicting PE from lipidomic measurements. This analysis was performed using the *mixOmics* package in R. We built an initial sPLS-DA model with 10 components using the lipidomic measurements as the predictor variables and the pregnancy group as the outcome variable. To choose the number of components to retain in the model, we assessed model performance when sequentially including additional components (Appendix A). Model performance was assessed using 10-fold cross validation repeated 10 times. We used the classification error rate, corrected for imbalances in group size, as a metric of model performance. The model which gave the lowest classification error rate was then selected for further fine tuning by variable selection. For each component, the number of variables retained was determined by varying the number of variables and subsequently assessing model performance using 10-fold cross validation repeated 10 times. The number of variables which gave the lowest classification error rate for each component was retained in the final tuned sPLS-DA model (Appendix A). After choosing the optimal number of components and variables, we tested the predictive performance of each model. To do this, we first randomly split data into training and test sets. Subsequently, we fit sPLS-DA models to the training and then tested their predictions on the training data. The detection rate was calculated as the fraction of total PE cases correctly classified. We repeated this process 10 times for each model to obtain an estimate of predictive performance.

## 3. Results

### 3.1. Maternal Demographics and Pregnancy Characteristics

Patients in this study included women with obesity who either had an uncomplicated pregnancy (CTL) or developed PE. Women who later developed PE entered the study at the same gestational ages as CTL and blood samples were obtained at similar gestational ages in the second and third trimester (Table 1). At entry to the study, maternal age, BMI, blood pressure before 20 weeks, smoking status, and racial distributions were not different between groups (Table 1). As expected, gestational age at delivery and birthweight percentile were lower and blood pressure higher in the PE group (Table 1). Blood pressures were not different in the two groups at admission, but diastolic pressures were slightly higher in second trimester in the women destined to develop PE (70 ± 8 mm Hg) compared to those who did not (67 ± 8 mm Hg, *p* = 0.05). In the third trimester, prior to the development of PE, women destined to develop PE had both higher diastolic (72 ± 8 vs. 67 ± 8 mm Hg, *p* = 0.04) and systolic pressure (119 ± 8 vs. 113 ± 10 mm Hg, *p* = 0.02). As expected, women with PE had higher systolic (146 ± 8 vs. 123 ± 10 mm Hg, *p* < 0.0001) and diastolic blood pressure (89 ± 8 vs. 71 ± 7 mm Hg, *p* < 0.0001) just prior to delivery.

### 3.2. Dynamics of Broad Lipid Classes during Pregnancy

Standard lipid panels revealed well-recognized changes in triglycerides, total cholesterol, HDL, and LDL across pregnancy (Appendix A) [25]. However, we found no differences in these variables between PE and CTL subjects at any trimester of pregnancy (Appendix A). Therefore, we performed mass spectrometry-based targeted plasma lipidomics on the Lipidyzer platform to identify lipid subclasses or individual lipid species that might distinguish PE and CTL groups.

We first assessed the dynamics of broad lipid classes afforded by plasma lipidomics (Figure 1). Cholesterol esters (CE) increased as pregnancy progressed in both groups (Figure 1A) consistent with the results of the standard lipid panel findings in Appendix A. Ceramide moieties (Figure 1B–D) and free fatty acids (FFA; Figure 1F) remained stable throughout pregnancy. Several lipid classes increased in both groups during pregnancy including diacylglycerols (DAG; Figure 1E), lysophosphatidylethanolamines (LPE; Figure 1H), phosphatidylcholines (PC; Figure 1I), phosphatidylethanolamines (PtdEtn; Figure 1J), plasmalogens (Figure 1K), and sphingomyelins (SM; Figure 1L). Lysophosphatidylcholine (LPCs; Figure 1G) levels decreased throughout the course of pregnancy in both groups. The only broad lipid class that differed between groups was plasmalogens which increased by 19% (*p* = 0.0153) in the third trimester of PE pregnancies compared to control pregnancies (Figure 1K). Third trimester plasmalogen levels were not correlated with time since last food in either group (Appendix A), indicating that these differences were not driven by recently ingested lipids.

### 3.3. Third Trimester Differences in Individual Lipid Species between Uncomplicated and Preeclamptic Pregnancies

Plasma lipidomics afforded measurements of 280 individual lipid species in CTL and PE groups at each trimester of pregnancy. We first probed for individual lipid species whose concentrations differed between groups by performing multiplicity adjusted (FDR < 0.1) Mann–Whitney U tests on data from each trimester. Surprisingly, there were no significant differences in any lipid species between groups during the first two trimesters. During the third trimester, however, there were nine lipid species that were significantly higher in PE patients (Figure 2A). We analyzed these nine lipids individually throughout the course of pregnancy using linear mixed effects models. Two PtdEtn lipids with ester fatty acid linkages, PtdEtn (18:2/16:1) and PtdEtn (18:1/22:4) were 33.8% and 22.4% higher in PE women, respectively (Figure 2B,C). Additionally, several plasmalogen PtdEtn lipids were significantly higher in PE pregnancies including 18:0/20:3 (40.5%, Figure 2D), 16:0/20:3 (41.3%, Figure 2E), 18:1/20:3 (34.7%, Figure 2F), and 18:0/20:4 (32.6%, Figure 2G). Three FFA species, 15:0 (18.8%, Figure 2I), 16:1 (25.6%, Figure 2H), and 18:3 (32.5%, Figure 2J), were also greater in the PE group. While some of these lipids were slightly higher in PE during the second trimester, they all only became significantly increased by the third trimester.

### 3.4. Exploratory Analysis of Factors Driving Plasma Lipidomic Variation

To investigate sources of plasma lipidomic variation between individuals in an unsupervised manner, we performed exploratory principal component analysis (PCA). Most lipidomic variation (53%) was explained by the first two principal components (Figure 3A). The first principal component, which accounted for 41% of variation, was largely defined positively by LPC species and negatively by PC species (Figure 3B). The second principal component (12% of variation) was defined positively by PtdEtns and negatively by various LPCs and FFAs (Figure 3B). Thus, phospholipids are the major source of plasma lipidomic differences between patients.

To visualize lipidomic variation between patient groups and trimesters of pregnancy, we projected samples onto the first two principal components (Figure 3C). These two components gave the most apparent visual separation of samples. There was no obvious clustering of samples (Figure 3C), indicating that plasma lipidomes were relatively homogeneous in this dataset. We did observe some separation of samples by trimester of pregnancy (Figure 3D). However, samples from CTL and PE groups were largely overlapping when combining data from all trimesters (Figure 3E). Thus, the trimester of pregnancy appears to be a larger contributor to lipidomic variability than if a woman develops PE later in pregnancy or not.

### 3.5. Lipidomic-Based Predictive Model of Pregnancy Outcome

We next utilized sparse partial least squares discriminant analysis (sPLS-DA) to determine whether plasma lipids can predict pregnancy outcome at various stages of pregnancy. sPLS-DA is a supervised multivariate classification technique which finds combinations of lipids, called components, that best predict pregnancy outcome. We built a separate sPLS-DA model for each trimester. Details regarding the selection of the number of components and lipids per component are shown in Appendix A. The PE detection rates for the first, second, and third trimester models were 20, 19, and 16%, respectively. For reference, the Fetal Medicine Foundation Model, which utilizes various maternal characteristics, uterine artery pulsatility index, mean arterial pressure, and placental growth factor levels, can predict 42.5% of all PE cases [26]. The lack of predictive power of our dataset can also be seen in Figure 4A–C where samples from both groups are largely overlapping when projected onto the first and second components of each sPLS-DA model. In summary, our plasma lipidomic measurements did not afford a strong predictive model of PE in this cohort of obese women.

### 3.6. Effects of Race and Fetal Sex on Plasma Lipids

We also assessed whether fetal sex or race influenced plasma lipids as measured by standard lipid panels. We found no significant differences in lipid levels between pregnancies with male or female fetuses (Appendix A). There were also no significant differences in lipids by PE status when stratifying pregnancies by fetal sex (Appendix A). Comparing patients by self-reported race, on the other hand, did reveal differences in lipid concentrations (Appendix A). Triglycerides were higher in White women than in Black women across all three trimesters of pregnancy (*p* < 0.0001). Total cholesterol was higher in White women than in Black women in the second (*p* = 0.0214) and third (*p* = 0.00021) trimester. Moreover, the change in total cholesterol across pregnancy was significantly different between White and Black women (*p* = 0.003). Black women who later developed PE had higher levels of trigylcerides throughout pregnancy than Black women with uncomplicated pregnancies (Appendix A). Given the small number of White women in our study, we were not powered to compare differences in lipid concentrations in White women who did or did not develop PE.

We also compared individual lipid species, as determined by LC-MS, between CTL and PE groups in each race individually using Mann–Whitney U tests. We only observed significant differences between CTL and PE groups in Black women during the third trimester (Appendix A). Black PE women had higher levels of several PtdEtn lipids (Appendix A), including some of the same plasmalogen species identified when comparing all CTL and PE patients (Figure 2). We also compared lipids between all Black and White women to determine whether the plasma lipidomic background on which PE develops is different between races. There were many lipid species that significantly varied between groups, the majority of which were decreased in Black women (Appendix A–D). Several species, including two DAG species, were consistently decreased in Black patients throughout pregnancy. Furthermore, we observed some separation between Black and White women during all three trimesters when using PCA to project their lipidomes into two-dimensional space (Appendix A–G). On the contrary, samples did not cluster by fetal sex (Appendix A), despite female fetal sex being a risk factor for PE [19]. These findings indicate that race, but not fetal sex, contributes to variability in the plasma lipidome of pregnant women with obesity.

## 4. Discussion

In this study, we performed plasma lipidomics throughout pregnancy to identify lipid species that might differ in obese women who later developed PE. While plasma lipidomes were quite similar between CTL and PE groups for the first two trimesters, we identified nine lipid species that were significantly elevated in the third trimester of PE pregnancies. These included PtdEtn plasmalogens, PtdEtns, and FFAs which may speak to the pathophysiology of PE.

Using targeted LC-MS, we identified four PtdEtn plasmalogen species whose concentrations were increased in the third trimester of PE pregnancies. These glycerophospholipids contain an ethanolamine head group with a vinyl ether-linked lipid at the *sn-1* position and ester-linked lipid at the *sn-2* position [27]. Plasmalogens are found both in membrane fractions in a variety of tissues as well as in the plasma bound to lipoproteins (mostly HDL) [28]. Their vinyl ether bond is susceptible to oxidation and thus plasmalogens are thought to be scavengers of reactive oxygen species [29,30]. Chronic disease states associated with oxidative stress, such as aging, neurodegeneration, and heart disease, are typically associated with reduced serum plasmalogens [31]. This is thought to reflect enhanced oxidation of plasmalogens, which consumes the plasmalogen and yields other lipid species. Conversely, obesity and elevated fasting blood glucose are associated with increased levels of plasmalogens, including three that we identified in the present study, PE(P-16:0/20:3), PE(P-18:0/20:4), and PE(P-18:0/20:3) [32,33,34,35,36].

It is unclear how elevated plasmalogens, as we found in our study, contribute to PE. PE has been associated with placental hypoperfusion which can result in intermittent hypoxia and enhanced generation of reactive oxygen species [37]. It is possible that the liver senses this oxidative stress and then increases the plasmalogen content of secreted lipoproteins [38] to mount a protective antioxidant response. As plasmalogen levels still remain elevated in PE, this suggests their secretion rate outpaces their oxidative consumption and/or clearance. Measuring plasmalogen synthesis and oxidative degradation, perhaps with lipid tracing techniques [39], could help make this determination. In addition to altered endogenous plasmalogen handling, it is also possible that elevated dietary intake of plasmalogen-containing foods, such as meat [40], contributes to increased plasma levels. Western diets, which are characterized by high meat intake, have been associated with increased risk for PE [41].

Another objective of our study was to determine whether differences in individual plasma lipid species might identify the 10% of women with obesity at risk for PE. Ideally, these lipid differences could be detected early in pregnancy, similar to other metabolomic biomarker panels that have been found in non-obese women [17,42]. However, we did not observe differences in any lipids during the first and second trimester. Furthermore, sPLS-DA classifier models at any trimester of pregnancy only correctly predicted 15–20% (detection rate) of PE cases. Thus, plasma lipidomics was unable to identify an early pregnancy biomarker panel that can predict PE in obese women.

There are several potential explanations for why we were unable to identify early pregnancy lipid biomarkers of PE in obese women. First, obesity-induced dyslipidemia may mask the early lipid changes that have been reported in non-obese PE subjects. Another possibility is that other plasma lipids, not measured by the Lipidyzer, may be early PE biomarkers in obese women [17,42]. For instance, eicosanoids are oxidized polyunsaturated fatty acid derivatives that have been heavily implicated in PE pathogenesis [43]. Recently, Watrous et al. used non-targeted mass spectrometry to detect over 500 of these lipid species in human plasma [44]. Thus, it may be necessary to expand the scope of detected lipids in order to identify biomarkers of PE in obese women. It is also possible that there are differences in lipid composition between various lipoprotein fractions that are lost when assaying whole plasma. Finally, because we used non-fasting samples in this study, we cannot rule out the possibility that recently ingested lipids may mask underlying lipid differences between groups. This seems unlikely, however, as there were no systematic differences between groups in time since last food at sampling.

In summary, we have utilized targeted plasma lipidomics to identify several lipid species, including plasmalogens, that are elevated prior to the development of PE in obese women. While plasma lipidomics did not afford an early trimester biomarker of PE in obese women, it did raise the possibility that plasmalogens may be involved in the pathogenesis of PE.

## Figures and Tables

**Figure 1 jcm-12-02970-f001:**
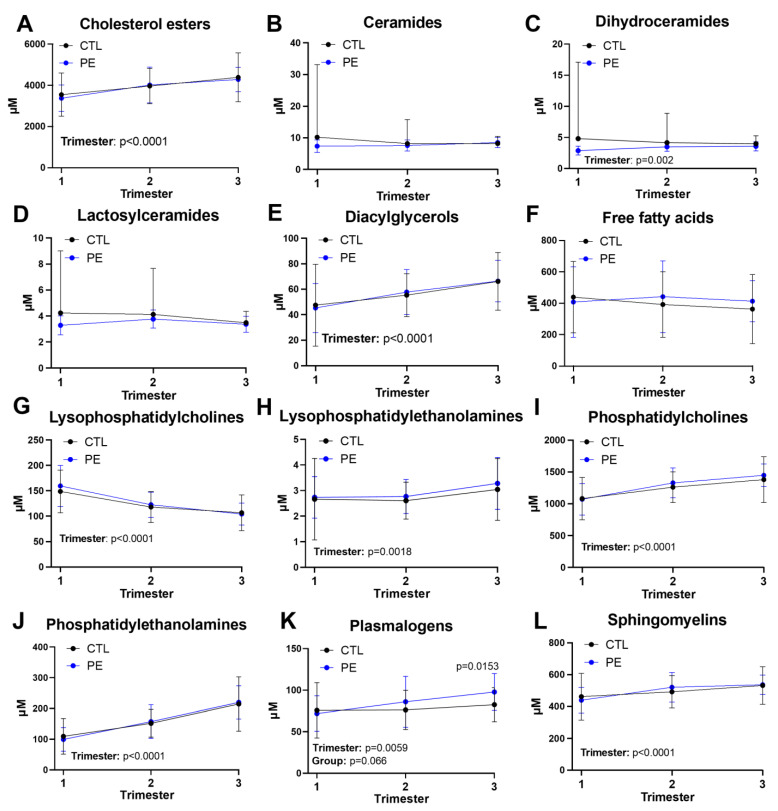
Changes in broad lipid classes during pregnancy. (**A**–**L**) Plasma concentrations of broad lipid classes at each trimester in uncomplicated (CTL) and preeclampsia (PE) pregnancies. Lipid class concentrations were calculated by summing the concentrations of all individual lipid species, measured by mass spectrometry, within a class. Significant sources of variation were determined by linear mixed effects modeling and are indicated in the lower left of each panel. Šídák’s multiple comparisons test was used to compare CTL and PE groups at each trimester. Error bars represent SD.

**Figure 2 jcm-12-02970-f002:**
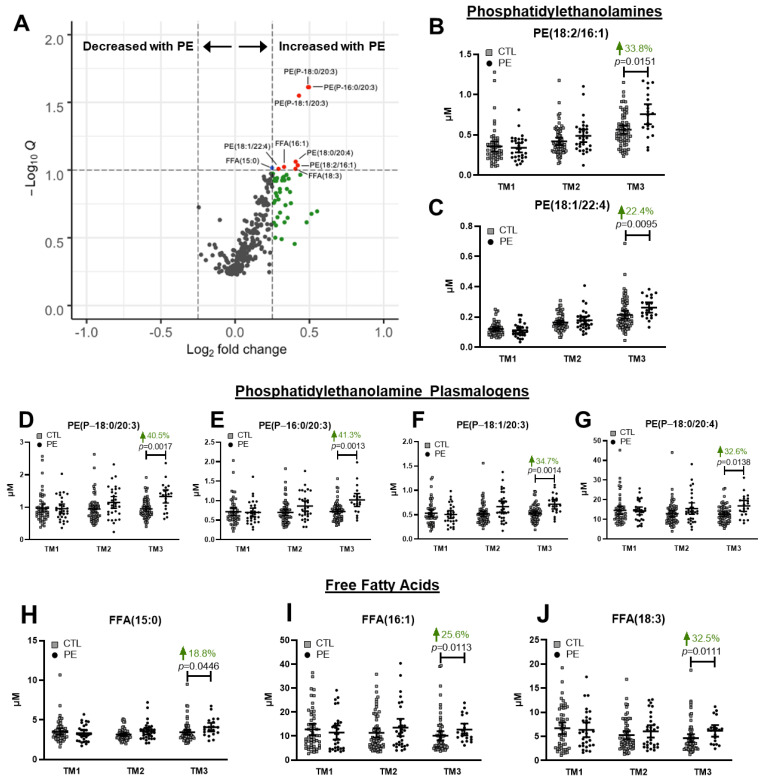
Individual lipid species elevated in the third trimester of PE pregnancies. (**A**) Volcano plot of 280 individual lipid species profiled in the third trimester of pregnancy. The x-axis indicates the log2 fold change relative to the CTL group. The y-axis shows the negative log 10 of the false discovery rate (q-value). Red points are lipid species with log2FC > 0.25 and q < 0.1. Blue points have q < 0.1. Green points have log2FC > 0.25. Lipid species with q < 0.1 are labeled. (**B**–**J**) Concentrations of individual lipid species (q < 0.1) in PE and CTL groups at each trimester of pregnancy. *p*-values are derived from Šídák’s multiple comparisons test following linear mixed effects modeling. Error bars represent 95% confidence intervals.

**Figure 3 jcm-12-02970-f003:**
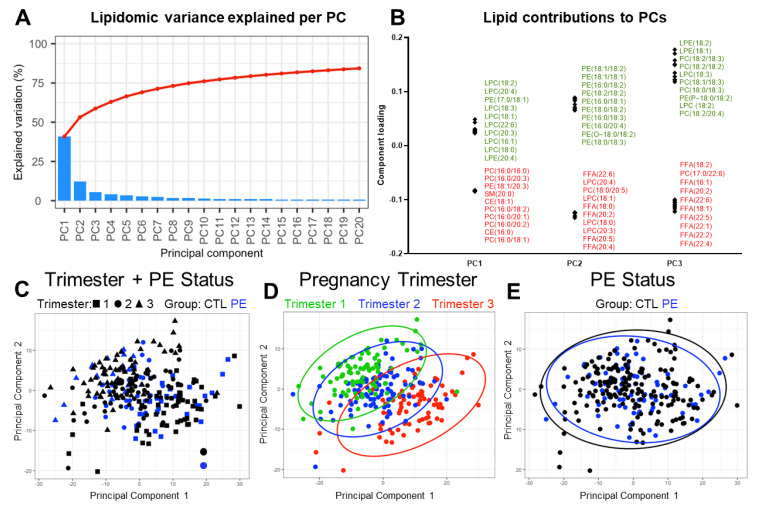
Unsupervised principal component analysis of plasma lipidomes from pregnant women with and without PE in the setting of obesity. (**A**) Scree plot showing the percent of lipidomic variance explained by each principal component. (**B**) Individual lipids which contribute most strongly to the first three principal components. The top 10 positive contributing lipids are shown in green and the top 10 negative lipids are shown in red. (**C**–**E**) Lipidomic data projected onto the first two principal components. Each data point represents a specific sample. Data are displayed by (**C**) trimester and group, (**D**) trimester, and (**E**) group. Ellipses indicate the 95% confidence ellipse of each group.

**Figure 4 jcm-12-02970-f004:**
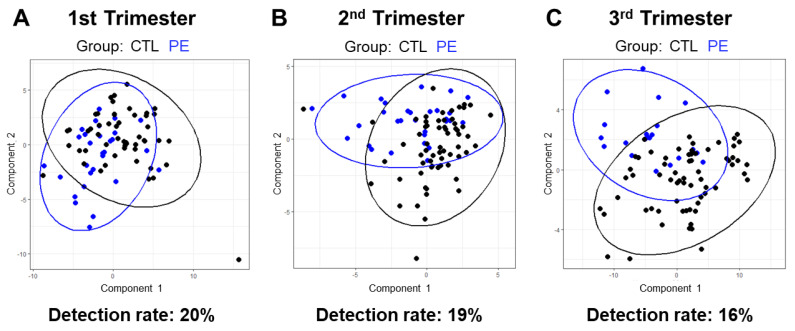
Lipidome-based PE prediction using supervised sparse partial least squares discriminant analysis. (**A**–**C**) Lipidomic data projected onto the first and second components of an sPLS-DA model built using data from each of the indicated trimesters. Ellipses indicate the 95% confidence interval of each group. The PE detection rate for each trimester-specific model is shown below its corresponding plot.

**Table 1 jcm-12-02970-t001:** Clinical Data of Cohort.

	Uncomplicated Pregnancy (n = 67)	Preeclampsia (n = 33)	*p*-Value
Age at delivery (years)	23 (21, 26)	24 (21, 27)	0.61
BMI pre-pregnancy (kg/m^2^)	36.2 (33.0, 41.2)	36.0 (32.1, 40.7)	0.67
Race (n,%)			
Black	44, 66%	22, 67%	0.92
White	22, 33%	9, 27%	0.57
Other	1, 1%	2, 6%	
Cigarette smokers (n,%)	26, 39%	15, 45%	0.53
Primiparous (n,%)	53, 73%	23, 70%	0.31
Gestational age wks. at sampling			
First trimester	8.9 ± 2.3 (n = 52)	8.1 ± 1.7 (n = 29)	0.28
Second trimester	19.8 ± 1.3 (n = 63)	19.7 ± 1.7 (n = 30)	0.43
Third trimester	34.9 ± 0.8 (n = 65)	35.1 ± 1.2 (n = 20)	0.51
Gestational age at delivery (wks)	39 (38, 40)	38 (35, 39)	<0.0001
Birth weight gm	3335 (3080, 3636)	2820 (2282, 3445)	<0.0001
Birth weight percentile ^1^	62 (36, 77)	30 (10, 59)	0.017
First trimester blood pressure ^2^			
Systolic (mmHg)	116.5 ± 8.3	116.2 ± 7.9	0.86
Diastolic (mmHg)	70.3 ± 6.8	73.3 ± 6.5	0.06
Second trimester blood pressure ^2^			
Systolic (mmHg)	113.3 ± 9.0	116.8 ± 8.6	0.073
Diastolic (mmHg)	66.8 ± 7.5	70.1 ± 7.9	0.049
Third trimester blood pressure ^2^			
Systolic (mmHg)	113.3 ± 10.2	119.0 ± 7.7	0.021
Diastolic (mmHg)	67.2 ± 8.4	71.5 ± 8.1	0.041
Pre-delivery blood pressure^2^			
Systolic (mmHg)	123 ± 8	146 ± 8	<0.0001
Diastolic (mmHg)	71 ± 7	89 ± 8	<0.0001
Infant sex (n,% female)	33, 49%	14, 42%	
Preterm delivery < 37 wks (n,%)	1, 1%	12, 36%	<0.0001
Severe preeclampsia (n,%)		12, 36%	
Gestational diabetes (n,%)	6, 9%	2, 6%	0.62

Data are shown as mean ± standard deviation unless data were not normally distributed, in which case data are shown as median (25th percentile, 75th percentile). Subjects were group matched for the first six clinical parameters. ^1^ Birth weight percentiles were calculated from data for births at Magee-Womens Hospital using both sex of the baby and race. ^2^ Trimester blood pressures were measured at study visits when blood samples were collected. Pre-delivery blood pressure was the average of clinical blood pressures at admission to labor and delivery and prior to treatments which alter blood pressure.

## Data Availability

Data are contained within the article or Appendix A.

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
