# Peer review of "Lipidomics Reveals Elevated Plasmalogens in Women with Obesity Who Develop Preeclampsia"

_jcm, 2023, doi:10.3390/jcm12082970_

Round 1

Reviewer 1 Report

Comments

Lines 35-36: “Preeclampsia (PE) is a common pregnancy disorder (3-5% of all pregnancies) that is a major cause of maternal and infant mortality worldwide [1].”

The reference cited in this paragraph describe that preeclampsia complicates between 2 - 8% of pregnancies, so it is suggested to support this paragraph with specific literature on the epidemiology of preeclampsia worldwide.

Line 40: Reference 3 is from 2007 and there is a meta-analysis by Liona Poon in 2022 about the treatment with aspirin (Prevention of preeclampsia with aspirin: doi:10.1016/j.ajog.2020.08.045).

Lines 48-50: In the cited study (reference 6) women with obesity were compared with women with BMI of 21 and not 25, furthermore, the risk of PE was observed to decrease in women with BMI ≥35.

In general, I suggest updating references and not modifying their meaning.

What is the rationale for using parametric tests to analyze lipid class levels and nonparametric tests for lipid species?

The clinical characteristics of the patients who developed preeclampsia should be described in more detail. Although the authors mention that the ACOG criteria were used, many details of the study population are missing, such as percentage of c-sections, treatment, week of PE diagnosis, proteinuria data, selection criteria of participants, etc. Could the authors explain why the blood pressure of women with PE in the first trimester did not reach 140 mmHg, when this is a criterion for PE according to the 2002 ACOG criteria?

Line 192-193. “Total cholesterol was higher in White women than in 192 Black women in the second and third trimester”, p value in missing

Lines 267-269: The analysis showed that the main contributor to lipidomic differences was the trimester of gestation and not PE, so why is the article focused on PE? Add to this the fact that these lipidomic measurements were not shown to be predictors of PE (lines 288-300).

In table 1: the kilogram symbol should be written with a lower-case k, eliminate "P" and "NS" and use only numerical values in the p-value column, percentages for preterm delivery and gestational diabetes are missing.

Lines 341-359. The first paragraph of discussion is actually a description of the results.

Lines 256-259: The value of the first component suggests high co-linearity, but the subsequent ones contribute very little, why not make a variable selection to improve the explained variance? What were the criteria for components selection?

Line 274: It is mentioned here that there are 3 components, but in the graphs and the rest of the text there are only 2.

Lines 314-315: What would be the criteria to determine if the model is able to discriminate the outcome?

Lines 373-376. “It should be noted, however, that we analyzed non-fasting blood samples in our study. Thus, we cannot rule out the possibility that dietary lipids mask underlying lipidomic differences between groups during early pregnancy”. It is important to mention in the main text that samples from non-fasting women were used. In addition, it is imperative to discuss the use of this type of sample, and not just mention it as a potential confounding variable.

Overall, the discussion is highly speculative and appears to be based not on the findings, but on the results of other studies.

The biological significance of the lipid classes that showed differential concentrations between the groups and their role in the pathophysiology of PE remains to be discussed.

It is unclear what is the basis for the hypothesis proposed in Lines 404-406 to explain the alterations in plasmalogen concentrations.

The summary far exceeds the actual scope of the study.

Author Response

Point 1: Lines 35-36: “Preeclampsia (PE) is a common pregnancy disorder (3-5% of all pregnancies) that is a major cause of maternal and infant mortality worldwide [1].”

The reference cited in this paragraph describe that preeclampsia complicates between 2 - 8% of pregnancies, so it is suggested to support this paragraph with specific literature on the epidemiology of preeclampsia worldwide.

Response 1: Thank you for pointing this out. We have now included a reference to a proper epidemiology study (Abalos et al. Eur J Obstet Gynecol Reprod Biol. 2013) which estimates the global prevalence of PE to be 2.7 to 8.2%. We have also updated this estimate in lines 37-38.

Point 2: Line 40: Reference 3 is from 2007 and there is a meta-analysis by Liona Poon in 2022 about the treatment with aspirin (Prevention of preeclampsia with aspirin: doi:10.1016/j.ajog.2020.08.045).

Response 2: We have updated our aspirin reference to Liona Poon’s 2022 meta-analysis (line 41). As low-dose aspirin is an inhibitor of cyclooxygenase-1, it is possible that aspirin may prevent PE by reducing levels of various prostanoid species. Unfortunately, the Lipidyzer platform used in this study was not configured to detect these lipid species. However, we have included these lipid species as a potential mediator of PE in the Discussion (lines 339 – 343).

Point 3: Lines 48-50: In the cited study (reference 6) women with obesity were compared with women with BMI of 21 and not 25, furthermore, the risk of PE was observed to decrease in women with BMI ≥35.

Response 3: We have updated the reference BMI to 21. Bodnar et al. (ref. 6) did not, however, find the risk of PE to decrease at BMIs ≥35. The adjusted odds ratio for PE risk at a BMI of 35 was 2.4 (1.8, 5.7 95% CI). The estimates for adjusted odd ratios for pre-eclampsia development at a BMI of 40 were unstable due to a lack of sufficient power, as stated by the authors. Nonetheless, the adjusted odds ratio at 40 BMI was still 2.3 (0.8, 6.4 95% CI) relative to women with a BMI of 21.

Point 4: In general, I suggest updating references and not modifying their meaning.

Response 4: Reviewer 1’s point is well taken. We have carefully looked through references to ensure previous literature is cited appropriately.

Point 5: What is the rationale for using parametric tests to analyze lipid class levels and nonparametric tests for lipid species?

Response 5: We thank Reviewer 1 for highlighting this discrepancy. We initially used FDR-controlled nonparametric testing of individual lipid species due to the non-normal distributions these data follow. After identifying species of interest (Figure 2A), we wanted to determine the effect of PE status, trimester, and their interaction on these lipids. To the best of our knowledge, however, there are no non-parametric methods to test the effects of two independent variables on the response (lipid). Thus, we log-transformed the data to make them normally distributed, and then performed linear mixed effects modeling with a random effect to account for the repeated measures on individual patients (Figure 2B-2J).

With respect to the lipid class data (Figure 1), we have now determined that they also follow non-normal distributions. Thus, we should have performed linear mixed-effects modeling on the log-transformed data instead of 2-way ANOVA. We have now reanalyzed the data in this manner and updated Figure 1.  

The results of linear mixed effects of modeling were very similar to 2-way ANOVA for these data. The only changes were 1) we uncovered an effect of pregnancy trimester on dihydroceramides (Fig 1C), 2) the p-value of the group effect for plasmalogens increased from p=0.046 to p=0.066 and 3) the p-value of the 3rd trimester difference in plasmalogens decreased from p=0.06 to p = 0.0153.

These changes are now described in the Methods section (lines 138-142) and the main text (lines 213-215).

Point 6: The clinical characteristics of the patients who developed preeclampsia should be described in more detail. Although the authors mention that the ACOG criteria were used, many details of the study population are missing, such as percentage of c-sections, treatment, week of PE diagnosis, proteinuria data, selection criteria of participants, etc. Could the authors explain why the blood pressure of women with PE in the first trimester did not reach 140 mmHg, when this is a criterion for PE according to the 2002 ACOG criteria?

Response 6:  We apologize for not presenting experimental design more clearly. The women in the PE group did not develop PE until after the third trimester samples were taken. In fact, samples from closer than two weeks to delivery were excluded. We have now clarified the timing of sample collection in relation to development of pre-eclampsia (lines 95-97).

All pre-eclampsia subjects met the 2002 ACOG criteria, which we have now described in a section of the Methods (“Pre-eclampsia definition”). Also, please note the “Pre-delivery blood pressure” in Table 1 which shows that the PE group reached diagnostic blood pressure.

We did not include Cesarean section rate or treatment data because these would have occurred after the final samples were collected and did not seem relevant to the study. Since these were “normal” obese women, none had proteinuria until preeclampsia was diagnosed, which was after last sample was collected. The diagnosis of preeclampsia was made based upon pre-2013 criteria for preeclampsia which demanded proteinuria, therefore when the women had preeclampsia (after the last reported data and samples), all had proteinuria.

It is impossible to know the date of the diagnosis of preeclampsia. Since women are seen weekly to every other week at the stage of pregnancy at which preeclampsia occurs, we have no idea when the diagnostic criteria first occurred during their time away from assessment. However, since the standard management of preeclampsia near term and usually also preterm is expeditious delivery, a surrogate for date of diagnosis is date of delivery. This is used in virtually all studies in the literature.

Point 7: Line 192-193. “Total cholesterol was higher in White women than in 192 Black women in the second and third trimester”, p value in missing

Response 7: We have now included the p values for these comparisons (lines 291-292).

Point 8: Lines 267-269: The analysis showed that the main contributor to lipidomic differences was the trimester of gestation and not PE, so why is the article focused on PE? Add to this the fact that these lipidomic measurements were not shown to be predictors of PE (lines 288-300).

Response 8: Our manuscript discusses pre-eclampsia prediction because that was one of the original goals of the study. One hypothesis we tested was that there are changes in the plasma lipidome in the first or second trimester of pregnancy that might predict pre-eclampsia development in obese women. That did not turn out to be the case. We want to be transparent with readers about the purpose of the study even though we disproved this hypothesis.

Another goal of the study, more exploratory in nature, was to identify lipids which may contribute to the pathogenesis of pre-eclampsia. As discussed extensively throughout the manuscript, we found 9 lipid species that become elevated in the third trimester of pregnancy of obese women who develop pre-eclampsia. We, and hopefully other researchers, will use these findings to explore mechanisms by which these species contribute to or result from pre-eclampsia.  

We have clarified these objectives at the end of the Introduction (lines 64-65).

Point 9: In table 1: the kilogram symbol should be written with a lower-case k, eliminate "P" and "NS" and use only numerical values in the p-value column, percentages for preterm delivery and gestational diabetes are missing.

Response 9: We thank Reviewer 1 for these improvements. We have updated Table 1 as suggested.

Point 10: Lines 341-359. The first paragraph of discussion is actually a description of the results.

Response 10: Reviewer 1 is correct. We have written a brief summary of the results to begin our Discussion section in order to re-orient the reader to the main findings that we will discuss below. We have now modified this summary to better align with the rest of the Discussion.

Point 11: Lines 256-259: The value of the first component suggests high co-linearity, but the subsequent ones contribute very little, why not make a variable selection to improve the explained variance? What were the criteria for components selection?

Response 11: Reviewer 1 refers to our exploratory principal component analysis (PCA) shown in Figure 3 and described in Section 3.4. The purpose of this analysis is to determine the groups of lipids (i.e. components) which drive most of the variation in lipidomes between subjects. That the first component drives most of the variation, as astutely noted by Reviewer 1, is exactly the type of insight we hoped to gain from this initial exploratory PCA.

In Figure 4 and Section 3.5, we go on to use sparse partial least squares discriminant analysis (sPLS-DA) to build component-based predictive models. Here we use both component and variable selection, as Reviewer 1 recommends. To choose the number of components for the model, we iteratively added components to the model, until the performance of the model plateaued (Supplemental Figure 1A, C, E). After determining a sufficient number of components, we iteratively changed the number of variables on each component until the performance of the model no longer improved (Supplemental Figure 1B, D, E). In this way, we chose the number of components and variables per component per each model.

In order to add clarity to our methods and their execution we have 1) included more detail regarding these different analyses in the main methods section, rather than the supplement 2) changed the titles of Sections 3.4 and 3.5 to describe their contents in a more intuitive manner and 3) been more explicit in describing the differences in these analyses in the main text (lines 247-248; 270-271). We hope that this will add clarity to the methods and execution of our multivariate analyses.

Point 12: Line 274: It is mentioned here that there are 3 components, but in the graphs and the rest of the text there are only 2.

Response 12: When performing principal component analysis, n-1 principal components can be derived from a dataset where n is the number of variables. So in this situation, we actually have 279 components (280 lipid species). The challenge with exploratory principal component analysis is to decide which of these components are meaningful. In Figure 3B, which Reviewer 1 is referring to, we show lipids which contribute to the top 3 principal components. We showed the third component because we found the grouping of FFA species which contribute negatively to the third component to be of interest. For the subsequent plotting in Figures 3C-3E, we only show 2 components because a) it is easier to visualize in 2D space and b) plotting on the 3rd dimension did not provide any more meaningful separation of the data.

We have now added a line (line 255) to the main text to clarify our use of various numbers of principal components in Figure 3.

Point 13: Lines 314-315: What would be the criteria to determine if the model is able to discriminate the outcome?

Response 13: There is no hard and fast criteria for whether an sPLS-DA model can accurately predict an outcome, just varying degrees of predictive performance. Our best sPLS-DA model detected 20% of PE cases. For reference, the Fetal Medicine Foundation model, which utilizes various maternal characteristics, mean arterial pressure, the uterine artery pulsatility index, and placental growth factor levels, can predict 42.5% of pre-eclampsia cases (Tan et al. Ultrasound Obstet Gynecol. 2018). We have now included this reference in the main text (lines 275-277) to help readers evaluate the predictive performance of our model.

Point 14: Lines 373-376. “It should be noted, however, that we analyzed non-fasting blood samples in our study. Thus, we cannot rule out the possibility that dietary lipids mask underlying lipidomic differences between groups during early pregnancy”. It is important to mention in the main text that samples from non-fasting women were used. In addition, it is imperative to discuss the use of this type of sample, and not just mention it as a potential confounding variable.

Response 14: We did not collect fasting samples because we did not want to place additional stress on the pregnant women in our study. We have now added that non-fasting blood samples were used in the Methods section (line 94). We have also mentioned specific lipids, mainly free fatty acid species, which may be particularly susceptible to dietary intake (lines 346-347).

Point 15: Overall, the discussion is highly speculative and appears to be based not on the findings, but on the results of other studies.

Response 15: The first 3 paragraphs of the Discussion are focused almost exclusively on the results. We do agree with Reviewer 1 that subsequent paragraphs (lines 384 – 428), which discuss individual lipid species that we identified, are speculative. The reason for this is that there is almost no literature on how these lipids relate to pre-eclampsia. Thus, a certain level of speculation is required to tie these lipids to pre-eclampsia. However, we can acknowledge that we may have overextended our hypotheses. We have now condensed this section and only mentioned broad potential mechanisms (i.e.,oxidative stress) by which these lipid species may relate to pre-eclampsia.

Point 16: The biological significance of the lipid classes that showed differential concentrations between the groups and their role in the pathophysiology of PE remains to be discussed.

Response 16: The only broad lipid class that showed any changes between pre-eclamptic and uncomplicated pregnancies was plasmalogens. We have now more specifically referred to this finding in our discussion section about plasmalogens (lines 315-330)

Point 17: It is unclear what is the basis for the hypothesis proposed in Lines 404-406 to explain the alterations in plasmalogen concentrations.

Response 17: This hypothesis aims to explain some seemingly discordant findings: 1) diseases associated with oxidative stress (e.g., aging, heart disease) are associated with reduced plasmalogens 2) pre-eclampsia is associated with increased oxidative stress but 3) we find increased plasmalogens in women who develop pre-eclampsia.

We hypothesize that, during pre-eclampsia, oxidative stress is sensed by the liver. The liver responds to this oxidative stress, in part, by increasing the plasmalogen content of secreted lipoproteins. The evidence for this hypothesis is that a) lipoproteins secreted from cultured rat hepatocytes contain plasmalogens (Vance Biochim Biophys Acta. 1990) b) plasmalogens are found in human plasma lipoproteins (Wiesner J Lipid Res 2009) and c) plasmalogens can scavenge reactive oxygen species (Felde Chem Phys Lipids 1995). What we do not know is the stimulus for plasmalogen secretion nor the plasmalogen secretion and oxidation rates in pre-eclampsia. We have now clarified this hypothesis and noted its limitations (lines 325-330).

Point 18: The summary far exceeds the actual scope of the study.

Response 18: We agree with Reviewer 1 and have refocused the summary on actual findings of the study and not hypotheses that may stem from it.

Reviewer 2 Report

The authors studied a cohort of women with obesity throughout pregnancy to identify lipid mediators and/or biomarkers of PE.  Both targeted lipidomics and standard lipid panels in blood samples were analyzed in each trimester. Individual lipid species were compared by PE status at each trimester, as well as by self-identified race and fetal sex. Found that the insufficient antioxidant response and metabolic dysfunction accompaned with PE in the context of obesity.The research topic is interesting, the methodology is scientific, and the results are presented appropriately. I have a few suggestions for the authors.
1. Please provide more details about statistical analysis procedures, such as Sparse partial least squares discriminant analysis, to help the reader better understand the analysis correctness. 
2. The authors discussed the association between obesity and preeclampisa in the introduction and discussion section. Some papers in high quality journals demonstrated their relationships, such as 'Preeclampsia Prevalence, Risk Factors, and Pregnancy Outcomes in Sweden and China',  2021. JAMA Netw Open, 4: e218401.  It would be better the authors cite the article. 
3. Please correct all the small errors such as spelling carefully.

Author Response

Point 1:  Please provide more details about statistical analysis procedures, such as Sparse partial least squares discriminant analysis, to help the reader better understand the analysis correctness. 

Response 1: We have now included more details on linear mixed effects modeling (lines 124-135 & lines 138-142), principal component analysis (lines 157-160), and sparse partial least squares discriminant analysis (lines 162-176) in the Methods Section.

Point 2: The authors discussed the association between obesity and preeclampisa in the introduction and discussion section. Some papers in high quality journals demonstrated their relationships, such as 'Preeclampsia Prevalence, Risk Factors, and Pregnancy Outcomes in Sweden and China',  2021. JAMA Netw Open, 4: e218401.  It would be better the authors cite the article.

Response 2: We thank Reviewer 2 for highlighting this paper. We have now referenced it in the Introduction (line 47).

Point 3: Please correct all the small errors such as spelling carefully.

Response 3: We have now gone through the manuscript meticulously to correct any errors.

Reviewer 3 Report

The authors studied in a cohort of obese women during pregnancy the existence of lipid markers or biomarkers that can be used in preeclampsia (PE). Blood samples were obtained in each trimester and analyzed by both targeted lipidomics and standard lipid panels. Individual lipid species were compared by PE status at each trimester, as well as by self-identified race (Black vs White) and fetal sex. The results show that the lipid panels used and the clinical measurements performed showed no differences between PE and uncomplicated pregnancies. But on the other hand, Targeted lipidomics identified plasmalogen, phosphatidylethanolamine, and free fatty acid species that were elevated in the third trimester of women with PE. In addition, they observed that race and the trimester of pregnancy in which the mother is found are responsible for variations in plasma lipid profiles in women with obesity. The conclusions drawn by these authors indicate that lipid species that have been associated with antioxidants and alterations in glucose homeostasis are elevated in obese mothers who develop PE. These results suggest that an insufficient antioxidant response and metabolic dysfunction accompany PE in the context of obesity.

I find interesting this approach that aims to see why a percentage of obese pregnant women develop preeclampsia, while another percentage does not.

Introduction

Although the main objective is well justified in the introduction, that is, the search for markers that allow us to know why a percentage of obese mothers develop PE and others do not, the reason for considering lipid profiles as one of these markers is not, in my opinion, well justified, especially when it indicates that no differences have been observed in other studies.

Furthermore, in the final paragraph of the introduction, it already indicates the results to be obtained and its conclusions, when what it should indicate is the objectives sought by this work. This paragraph does not make sense and should be corrected.

In addition, part of the results focus on a large variety of individual lipid species determined, but there is no justification for such a large number. Would it not have been better to focus on a few lipid classes that could affect PE more? At the very least, the need for these determinations should be justified.

Material and Methods

It is not clear how the study population was selected from the initial population, i.e., since only 100 (33 with PE and 67 with normal gestation) of 600 obese pregnant women were used, why were obese pregnant women with a BMI equal to 30 kg/m2 not considered?. This point should be clarified a little more.

I do not understand why a section is made to describe the statistics, but only of the data and then continues with the determinations (lipodomics). I think it is better to first indicate all the determinations that are going to be performed and how they are going to be performed and then the statistics that are going to be carried out with the results obtained. This should be modified.

On the other hand, the material and methods is difficult to follow if we consider both, the material and methods of the article and the supplementary material. There are aspects that are easy to identify where they go, but others are not.

Why is the sample collection in the supplementary material and not in the article?

Why is it not indicated in the article which parts of the methodology are going to be expanded in the supplementary material? supposedly, this material is only to add information of a complex methodology with abundant information and that in order not to expand too much the size of the article it is given as supplementary material, but it should be indicated in the text of the article to facilitate the understanding of the methodology and besides, it should only be an expansion of methods described in the article, not part of the methodology not described in the article.

Results

I do not understand the percentages given in Table 1 in relation to race. Are they in relation to the total number of individuals in each group, in relation to the total number of individuals in general, in any case the sum of percentages does not fit me. In addition, because this variable does not indicate the value of p.

The p value for the variable multiparous or preterm deliveries is also not indicated.

Why has the variable sex and race been considered only to see if it can affect the lipid profile and no other variables, such as blood pressure, number of deliveries or prematurity.

Why do they consider race, if they then indicate that there are not enough white pregnant women to be able to carry out the comparison?

In addition, they indicate that in black pregnant women they do not find differences between PE and uncomplicated gestation, however, in table S3, the TG do show significant differences (0.03).

In the results concerning individual lipid species, it is indicated that 286 are shown, but in the material and methods it is indicated that 460 individual lipid species are identified and data are given for 385. Furthermore, the internal standards (indicated in the supplementary material) speak of 54. I believe that this number of data should be clarified.

The final data related to individual lipid classes are really difficult to follow since it mixes graphs that compare control with PE with graphs that show general evolution during pregnancy, it talks about principal components, but it does not indicate which lines are these principal components, that is, a lot of data that are difficult to follow.

I really believe that the article would be much better if it focused on a few aspects and did not want to cover so much, for example, lipid differences between pregnant women with preeclampsia and those without preeclampsia, without considering variables that have no significant differences between the groups.

Discusión

As I have commented before, if one were to focus on the data indicated in the first paragraph of the discussion: “While pre-pregnancy obesity triples the risk for PE, only 10% of women with obesity develop PE [10]. It is unclear how these women, despite elevated risk, are able to avoid PE. Given that obesity causes dyslipidemia, it is possible that alterations in lipid profiles might drive different pregnancy outcomes. At the level of standard lipid panels, however, we did not observe any differences between CTL and PE pregnancies at any point during pregnancy. Therefore, we used a highly multiplexed targeted LC-MS strategy to identify individual lipid species that might vary between women with obesity who had uncomplicated pregnancies or developed PE. While plasma lipidomes were quite similar between CTL and PE groups for the first 2 trimesters, we identified 9 lipid species that were significantly elevated in the third trimester of PE pregnancies. These included PtdEtn, PtdEtn plasmalogen, and FFA species which may speak to the pathophysiology of PE.” It would still be interesting and easier to follow. Other aspects such as sex or race, among others, provide data, but perhaps it would be more convenient to introduce them in another publication.

Finally, it is not clear why the differences observed in a small number of individual lipid classes (9 out of almost 460 determined) could affect the obese pregnant woman to develop preeclampsia. Furthermore, if the major differences are seen in relation to plasmalogens (both in total and in individual classes), it would have been interesting to focus the discussion more on these data (relationship with the cardiovascular system, gestational diabetes, hypertension, etc.).

In general, I consider that the article is interesting in its approach or in its main objective, but that it is difficult to follow, both in its methodology and in its results, and furthermore, some determinations are not well justified, nor are some of the most interesting results discussed.

I believe that an excessive amount of data is given and most of them do not contribute anything to the main objective, they are only complements to the main results.

Author Response

Point 1: Although the main objective is well justified in the introduction, that is, the search for markers that allow us to know why a percentage of obese mothers develop PE and others do not, the reason for considering lipid profiles as one of these markers is not, in my opinion, well justified, especially when it indicates that no differences have been observed in other studies.

Response 1: While broad lipid classes (triglycerides, lipoproteins, etc.) may not be altered in obese women prior to pre-eclampsia, it is possible that individual lipid species are. This seems to be the case in lean women. Namely, Lee et al. found several phospholipid species that were elevated in lean women at 16-24 weeks of pregnancy and predicted pre-eclampsia (Lee et al. Sci Rep 2020). We wanted to test whether this would also be true in obese women. We have now added this reference to our justification for using lipidomics in the Introduction (lines 58-59).

Point 2: Furthermore, in the final paragraph of the introduction, it already indicates the results to be obtained and its conclusions, when what it should indicate is the objectives sought by this work. This paragraph does not make sense and should be corrected.

Response 2: We have modified the final paragraph of our Introduction to focus on the objectives of the study (lines 62-65).

Point 3: In addition, part of the results focus on a large variety of individual lipid species determined, but there is no justification for such a large number. Would it not have been better to focus on a few lipid classes that could affect PE more? At the very least, the need for these determinations should be justified.

Response 3: This is the first longitudinal study of individual plasma lipid species in obese pregnant women who later developed PE. As such, we did not know a priori which lipid classes would be most relevant to PE. Therefore, we used lipidomics to cast as wide a net as possible to increase our chances of finding lipids that may differ between groups. We have explained this approach in the final paragraph of the Introduction (lines 63-64).

Point 4: It is not clear how the study population was selected from the initial population, i.e., since only 100 (33 with PE and 67 with normal gestation) of 600 obese pregnant women were used, why were obese pregnant women with a BMI equal to 30 kg/m2 not considered?. This point should be clarified a little more.

Response 4: From the initial cohort of 600 women who had a BMI > 30 kg/m2 (WHO definition of obesity), we selected all women that a) developed pre-eclampsia and b) had available blood samples for each trimester. This group included 33 subjects. We then selected 67 obese women who had uncomplicated pregnancies as controls. These control subjects were matched with the women who later developed preeclampsia for BMI, race, parity, gestational age at sampling and smoking status as our control group. We have clarified our study population in the Materials and Methods section (lines 68-80).

Women with BMI equal to 30 kg/m2 (and lower) were not included in this study because the WHO definition is > 30 kg/m2.

Point 5: I do not understand why a section is made to describe the statistics, but only of the data and then continues with the determinations (lipodomics). I think it is better to first indicate all the determinations that are going to be performed and how they are going to be performed and then the statistics that are going to be carried out with the results obtained. This should be modified.

Response 5: The purpose was to put the statistical analysis next to its corresponding data collection method. However, we have now modified the Materials and Methods section to group the data collection and statistical analysis methods.

Point 6: On the other hand, the material and methods is difficult to follow if we consider both, the material and methods of the article and the supplementary material. There are aspects that are easy to identify where they go, but others are not.

Response 6: We agree with Reviewer 3. We have now put all methods description in the main manuscript. We hope this makes it easier to follow our data collection and analysis.

Point 7: Why is the sample collection in the supplementary material and not in the article?

Response 7: As described for point 6 above, we have now moved the sample collection description to the main manuscript.

Point 8: Why is it not indicated in the article which parts of the methodology are going to be expanded in the supplementary material? supposedly, this material is only to add information of a complex methodology with abundant information and that in order not to expand too much the size of the article it is given as supplementary material, but it should be indicated in the text of the article to facilitate the understanding of the methodology and besides, it should only be an expansion of methods described in the article, not part of the methodology not described in the article.

Response 8: We agree with Reviewer 3, it can be confusing to go back and forth between the main manuscript and the supplement. As discussed above, we have moved all the methods to the main manuscript. We hope the continuity of methods will ease in their understanding.

Point 9: I do not understand the percentages given in Table 1 in relation to race. Are they in relation to the total number of individuals in each group, in relation to the total number of individuals in general, in any case the sum of percentages does not fit me. In addition, because this variable does not indicate the value of p.

Response 9: The percentages of each race are in relation to the number of individuals in each group. For instance, 44 of the 67 uncomplicated pregnancies were from Black women (66%). For each group, uncomplicated pregnancy (66%+33%+1%) or pre-eclampsia (67%+27%+6%), all of the percentages add up to the expected 100%. We have now indicated p-values for the race comparisons between groups.

Point 10: The p value for the variable multiparous or preterm deliveries is also not indicated.

Response 10: We have now included the p-value for preterm delivery (p<0.0001) and primiparity (p=0.31) in Table 1.

Point 11: Why do they consider race, if they then indicate that there are not enough white pregnant women to be able to carry out the comparison?

Response 11: Even though we were underpowered to make comparisons between groups within White women, we were still sufficiently powered to make other meaningful comparisons related to race. Namely, we compared PE and uncomplicated pregnancies in Black women as well as the plasma lipidome of all Black women to all White women. We decided to make these comparisons because biomarkers and mechanisms of PE may vary depending on race. Indeed, we found several differences between Black and White women, as described in lines 290-294 and lines 299 - 307. It is imperative to understand these differences so that future PE diagnostics and treatments can be efficacious for all races.

Point 12: In addition, they indicate that in black pregnant women they do not find differences between PE and uncomplicated gestation, however, in table S3, the TG do show significant differences (0.03).

Response 12: We thank Reviewer 3 for pointing out this mistake. We have now indicated in the main text (lines 294-295) that Black women who go on to develop PE have higher triglycerides than those with uncomplicated pregnancies.

Point 13: In the results concerning individual lipid species, it is indicated that 286 are shown, but in the material and methods it is indicated that 460 individual lipid species are identified and data are given for 385. Furthermore, the internal standards (indicated in the supplementary material) speak of 54. I believe that this number of data should be clarified.

Response 13: We agree this can be clarified. The Lipidyzer mass spectrometry platform detected 460 individual lipid species. Of those 460, we only kept lipids that were detected on at least 4 of 5 mass spectrometer run days. The purpose of this is to remove lipids that are sensitive to daily technical variability. This triage left 385 lipids for analysis. Finally, we excluded lipids if they were undetected in more than 20% of samples in both groups at each time point. This left 280 final lipids for downstream analyses. We have now included this detail in the Materials and Methods (lines 116-119 and lines 146-149). 

As for the 54 internal standards, these cover a range of headgroups, acyl chains, and degrees of unsaturation that are found in the plasma lipidome. This allows for the accurate quantification of lipids detected by mass spectrometry. We have now included this detail in the Materials and Methods section (lines 108-111).

Point 14: The final data related to individual lipid classes are really difficult to follow since it mixes graphs that compare control with PE with graphs that show general evolution during pregnancy, it talks about principal components, but it does not indicate which lines are these principal components, that is, a lot of data that are difficult to follow.

Response 14: We presume that Reviewer 3 is talking about Figure 3 which describes the principal component analysis. We have tried to make this figure clearer in two ways. 1) We have labeled the graphs 3E – 3C with the comparison they show – i.e., combined, trimester effects, and pre-eclampsia status. 2) The principal components are the axes of these graphs. We have changed the axis label to read “Principal Component” instead of “PC.” We hope this makes the figure more easily interpretable.

Point 15: I really believe that the article would be much better if it focused on a few aspects and did not want to cover so much, for example, lipid differences between pregnant women with preeclampsia and those without preeclampsia, without considering variables that have no significant differences between the groups.

Response 15: We appreciate Reviewer 3’s perspective, although we respectfully disagree. We decided to include race as a variable because of the reasons discussed in Response 11. Namely, 1) there were differences in lipids between Black and White women and 2) race-driven differences in plasma lipidomics are important to understand. With respect to fetal sex, we thought this would be a variable of interest to readers because male fetal sex is a risk factor for pre-eclampsia.

Nonetheless, we can appreciate that the subgroup analyses may distract from the main comparison of the manuscript. Therefore, we have moved Figure 5, which shows the effect of race on individual lipids, to the Supplemental Data (Supplemental Figure 2). Furthermore, we have moved all description of the subgroup analyses in a single section (3.6: “Effects of race and fetal sex on plasma lipids”) at the end of the Results section.

We hope that this helps keep the manuscript more focused while also providing meaningful subgroup results to interested readers.

Point 16: As I have commented before, if one were to focus on the data indicated in the first paragraph of the discussion… ( It would still be interesting and easier to follow. Other aspects such as sex or race, among others, provide data, but perhaps it would be more convenient to introduce them in another publication.

Response 16: As discussed in response to Point 15, we have altered the manuscript to focus on the main comparison: women who develop pre-eclampsia versus those who have uncomplicated pregnancies. We have moved race and fetal sex data to the supplement and limited discussion to a single section at the end of the results.

Point 17: Finally, it is not clear why the differences observed in a small number of individual lipid classes (9 out of almost 460 determined) could affect the obese pregnant woman to develop preeclampsia. Furthermore, if the major differences are seen in relation to plasmalogens (both in total and in individual classes), it would have been interesting to focus the discussion more on these data (relationship with the cardiovascular system, gestational diabetes, hypertension, etc.).

Response 17: We agree with Reviewer 3 that the discussion should focus more on plasmalogens and their potential role in PE. We have now refocused our discussion of plasmalogens to highlight this finding (lines 315-330). We discuss the association of plasmalogens with other diseases, as well as our hypothesis for how they might be involved in pre-eclampsia.

Point 18: In general, I consider that the article is interesting in its approach or in its main objective, but that it is difficult to follow, both in its methodology and in its results, and furthermore, some determinations are not well justified, nor are some of the most interesting results discussed.

Response 18: We have made substantial changes to improve our manuscript as recommended by Reviewer 3. Namely, we have 1) included all methodological details to the main manuscript 2) moved subgroup analyses to the supplement to focus the manuscript on our main comparison 3) clarified our rationale for using lipidomics and 4) re-written the discussion to focus on differences in plasmalogens, our main finding. We hope that these changes have made our manuscript more focused and easier to follow.

Point 19: I believe that an excessive amount of data is given and most of them do not contribute anything to the main objective, they are only complements to the main results.

Response 19: While we agree with Reviewer 3 that our presentation of the results may distract from the study’s main objective, we maintain that the subgroup analyses will be of interest to readers. Thus, we have 1) moved all subgroup analyses to the supplementary data and 2) isolated our discussion of subgroup analyses to a single section at the end of the Results section.

Round 2

Reviewer 1 Report

Although the work done by the authors in this review process improved the presentation of the work, the fact that the samples were taken in non-fasting conditions represents a serious methodological issue that from my point of view does not allow any strong conclusion about, first, the increase of plasmalogens and, second, their involvement in preeclampsia.

Author Response

Point 1: Although the work done by the authors in this review process improved the presentation of the work, the fact that the samples were taken in non-fasting conditions represents a serious methodological issue that from my point of view does not allow any strong conclusion about, first, the increase of plasmalogens and, second, their involvement in preeclampsia.

Response 1: We appreciate Reviewer 1’s point that the use of non-fasting samples may confound our interpretation. Presumably, the concern is that recently ingested lipids may drive the elevated plasmalogens we see in the PE group. To address this, we have now included ‘time since last food’ data which was collected during our study. These data are shown in Supplemental Figure 1. We did not observe a difference in time since last food between groups at any trimester (Supplemental Figure 1A-C). Furthermore, there was no correlation between plasmalogen levels and time to feed in either group during the third trimester (Supplemental Figure 1D). These data strongly suggest that recently ingested lipids do not drive the lipid differences we observe in our study. We have included a discussion of this data in the methods (lines 99-100), results (lines 220-222), and discussion (lines 355-358).

While recently ingested food does not appear to be a confounding variable in our study, it is possible that chronic consumption of plasmalogen-containing foods may contribute to the increased plasmalogens in PE patients. The highest levels of ethanolamine plasmalogens are found in beef, pork, lamb, and chicken (Wu et al. Foods 2021). Thus, it is possible that elevated plasmalogens reflect increased intake of these foods. Western diets, which are characterized by higher meat intake, have been associated with increased risk for PE (Ikem et al. BJOG 2019). We have now included this hypothesis for elevated plasmalogens in the discussion (lines 338-341). 

Reviewer 3 Report

I consider that the authors have made multiple changes in the article, which has improved, in my opinion, both its follow up and understanding, which has increased its scientific contribution to this topic, which as I had commented I think was interesting.

However, there are still some aspects, let's say of lesser weight, that could be considered.

My main consideration, as I commented in the first review, focuses on the large amount of data contributed by the authors without doubt of scientific importance, but without a previous justification for their appearance. I refer to the data related to the issues of race and sex, they are secondary to the main objective, which alone, has sufficient data to support this publication.

However, after reading the responses of the authors, I consider that really, this contribution of race and sex, improves the publication and the scientific contribution of the same.

But perhaps it would be interesting if this aspect were indicated in the introduction, as a, let us say, secondary objective of the study and justify why this differentiation is going to be made. Thus, it would be better understood why, although with the limitation of the number of white mothers, this differentiation is made.

I really think that this point should be indicated in the introduction.

There are some other minor aspects.

For example, in relation to the selection of mothers with a BMI greater than 30, it should be pointed out that although the authors may have decided to consider only those mothers with a BMI greater than 30 to be included in the study, it is not entirely certain that this is because of the definition or the index given by the WHO, since, in adults, WHO considers obesity when the BMI is equal to or greater than 30. Although I really do not know if considering a BMI=30 would have increased the number excessively. It is true that the two reasons mentioned in this answer should be indicated in the text.

I really consider that the article has been improved, but also that more visualization should be given in the introduction to the topic of race and sex.

Author Response

Point 1: However, after reading the responses of the authors, I consider that really, this contribution of race and sex, improves the publication and the scientific contribution of the same.

But perhaps it would be interesting if this aspect were indicated in the introduction, as a, let us say, secondary objective of the study and justify why this differentiation is going to be made. Thus, it would be better understood why, although with the limitation of the number of white mothers, this differentiation is made.

I really think that this point should be indicated in the introduction.

Response 1: We thank Reviewer 3 for this suggestion. We have now included a justification for studying race and fetal sex in the Introduction and added these analyses as a secondary objective of the study (lines 65-68).

Point 2: For example, in relation to the selection of mothers with a BMI greater than 30, it should be pointed out that although the authors may have decided to consider only those mothers with a BMI greater than 30 to be included in the study, it is not entirely certain that this is because of the definition or the index given by the WHO, since, in adults, WHO considers obesity when the BMI is equal to or greater than 30. Although I really do not know if considering a BMI=30 would have increased the number excessively. It is true that the two reasons mentioned in this answer should be indicated in the text.

Response 2: Reviewer 3 is correct. The WHO considers adults with a BMI equal to or greater than 30 to be obese. Upon reviewing our BMI data, we do include patients that have BMIs of 30.0 and above. We apologize for this oversight. We have now corrected our methods section to reflect the inclusion of patients with a BMI of 30.0 as defined by the WHO (line 79).

Point 3: I really consider that the article has been improved, but also that more visualization should be given in the introduction to the topic of race and sex.

Response 3: We have now alluded to the race and fetal sex subgroup analysis in the Introduction as described above in Response 1.